# Mechanical Properties, Curing Mechanism, and Microscopic Experimental Study of Polypropylene Fiber Coordinated Fly Ash Modified Cement–Silty Soil

**DOI:** 10.3390/ma14185441

**Published:** 2021-09-20

**Authors:** Linfang Lu, Qiang Ma, Jing Hu, Qingfu Li

**Affiliations:** 1Construction Administration of The Second Phase Project of Zhaokou Irrigation District of the Yellow River, Kaifeng 475000, China; zkeqllf@163.com (L.L.); maqiangluhun@126.com (Q.M.); 2Institute of Rock and Soil Mechanics, Chinese Academy of Sciences, Wuhan 430071, China; 3School of Water Conservancy Engineering, Zhengzhou University, Zhengzhou 450001, China

**Keywords:** silty soil, polypropylene fiber, fly ash, UCS, SEM, XRD, curing mechanism

## Abstract

Silty soil has the characteristics of low natural moisture content and poor viscosity, and the strength and deformation required for foundation engineering can be satisfied by reinforcing and improving the silt. In order to study the reinforcement and improvement effects of polypropylene (PP) fiber and fly ash (FA) on cement–silty soil, an unconfined compressive strength (UCS) test, scanning electron microscope (SEM) test, and X-ray diffraction (XRD) analysis test were carried out. Cement (mixed amounts are 4%, 8%, 12%, and 16% of dry soil mass) was used as the basic modifier, and PP fiber (mixed amounts are 0%, 0.15%, 0.3%, and 0.45% of dry soil mass) compounded with FA (adding amounts of 0%, 5%, 10%, and 15% of dry soil mass) were used as an external admixture of cement–silty soil to study the mechanical properties, curing mechanism, and microstructure of the modified soil in different ages of 7 d, 14 d, 28 d, and 60 d. The test results show that with the increase in cement and curing age, the UCS of the modified soil increases, and with the increase in the PP fiber and FA, the UCS of the modified soil first increases and then decreases; there is an optimal content of FA and PP fiber, which are 10 and 0.15%, respectively. A large amount of C-S-H and AFt substances are produced inside the modified soil to cover the surface of soil particles or fill in the pores between soil particles, forming a tight spatial network structure and improving the mechanical properties of the cement–soil. The intensity of the diffraction peaks of the mineral components within the modified soils is more influenced by the cement and age, and the effect of FA is weaker. The stress–strain curve of the modified soil is divided into elastic stage, plastic deformation stage, and strain-softening stage, and the specimens in each stage have corresponding deformation characteristics. By analyzing the behavioral characteristics and curing improvement mechanism of modified soil from the duo perspective of macro-mechanical properties and microstructural composition, it can provide some basis for the engineering application of silty soil.

## 1. Introduction

Silty soil is formed by the long-term weathering of rocks, and its particles are mainly powder and sand particles, so the particles are poorly cohesive and have a loose texture, with low strength, poor grading, and strong capillary action [1,2,3]. In the eastern Henan Province of China, due to it is location in the middle and lower reaches of the Yellow River, the rivers in the territory constantly impact on the coastal areas, gradually forming a wide alluvial plain, and there are a large number of layered silty soils in the area. Moreover, the area belongs to the plain area and there is a relative lack of stone materials; it is not economical to replace the silty soil, so it is more practical to carry out reinforcement and improvement treatment.

At present, cement, lime, and other cementitious materials are commonly used in engineering to improve the treatment of silty soil or sandy soil [4], and the most commonly used is cement–soil, which is a composite material obtained by mixing the appropriate amount of cement and water with soil as the main component, with the advantages being that it is cheap and easy to obtain, has convenient construction, and can significantly improve the strength of raw materials. However, only adding cement to the soil can increase the strength of the soil, but the cement–soil is susceptible to high temperature, brittleness, and corrosion, and its structural stability can be damaged to a certain extent. Moreover, the large-scale use of cement not only causes high cost and waste of resources, but it also emits a large amount of carbon dioxide and dust when producing cement, which easily leads to the greenhouse effect and air pollution [5]. In addition, cement dust can affect the acidity and alkalinity of the soil, which in turn can cause crop yield reduction and other hazards [6]. In conclusion, finding other environmentally friendly materials to replace some of the cement and improving the engineering properties of plain cement–soils are the desirable strategies to improve the utilization of poor soils.

FA is the main waste product of coal-fired power generation, which is more harmful to the environment and can pollute groundwater and soil if disposed of by backfilling [7]. Mohanty et al. [8,9,10] found through their study that fly ash-modified soil has the characteristics of low early strength and poor water stability, and a single external admixture exposed many problems, but together with cement, these phenomena can be alleviated. In recent years, polymeric synthetic materials have been favored by the building materials market for their superior properties and low prices, of which fibers are one of the typical representatives. Synthetic fibers have high tensile strength and modulus of elasticity and can be added to the soil to limit the deformation of the soil, that is, “fiber-reinforced soil” in a broad sense. The construction steps of fiber-reinforced soil are simple and have a certain improvement effect on soft soil, expansive soil, silt, and other soils with poor engineering properties [11,12]. Fiber-reinforced soil belongs to physical reinforcement. Combining the physical and chemical properties of fly ash to improve cement–soil is a green and environmentally friendly method.

Scholars at home and abroad have conducted relevant studies around the improvement of silt or other types of soils. Xing et al. [13] studied the macro-mechanical properties and micro-composition structure of salt-rich cemented soils, and the test results showed that excessive cement content would reduce the strength of the cemented soils, and the differences in ion content led to more obvious differences in the microstructure of the cemented soils. Consoli et al. [14] studied the effects of moisture content, cement content, and porosity on the UCS of silt. The test results show that when the moisture content is constant, the strength and the cement content increase linearly. As the porosity increases, the strength of cement-modified silt decreases. Wei et al. [15] studied the mechanical properties of the soil after adding wheat straw, rice straw, jute, and polypropylene fibers to lime soil, respectively. The results showed that these four fibers can improve the compressive strength of lime soil, with polypropylene fibers having the best reinforcement effect, with the optimum fiber incorporation content of 0.2 or 0.25% and the optimum fiber length of 30 or 40% of the sample diameter. Boutouba et al. [16] explored the effect of cement on the mechanical behavior of sandy soil and investigated the shear strength characteristics and vertical deformation characteristics of cement–soil by direct shear tests. The test results showed that cement not only significantly improved the cohesion and internal friction angle of the sandy soil but also enhanced its ability to resist deformation, and the best improvement effect was achieved when the cement content was 10%. Liu et al. [17] studied the effects of the microstructure parameters of cement-modified roadbed silt on its macro-mechanical properties under freeze–thaw cycles. Through a static triaxial test and scanning electron microscopy test, it is found that as the cement content increases from 0% to 2%, the shear strength and shear strength parameters of the cement–soil increase by a factor of two, and the cohesion of the vegetal powder soil was more sensitive to the particle diameter, while the internal friction angle was more sensitive to the average particle abundance. Lo et al. [18] used cement and fly ash to reinforce silty soil and investigated the shear properties of the composite soil by triaxial test and observed the effect of particle bonding on strength and stiffness using a zero effective stress test. The test results showed that the initial expansion coefficient of the composite soil was lower than that of the plain soil, but the final expansion coefficient was high. The shear strength of the composite soil changes to a curved damage surface, which coincides with the plain soil under high stresses, and the damage function is established based on this. With the increase in confining pressure, the true bond stress and critical expansion rate of composite soil decrease, but the rate is different. Singh et al. [19] conducted a California bearing ratio (CBR) test on red soil and river embankment sand modified with low-calcium fly ash and cement. The results show that the CBR value of the two soil–ash mixtures immersed in water was higher than that of the unsoaked specimens, but the cement–ash mixtures showed the opposite trend. Both types of soils modified with 1% cement and 50% fly ash could be used as subgrade. Xiao et al. [20] mixed siliceous fly ash and cement into marine clay. Through a compression test and split tensile test, they found that the short-term strength growth rate of fly ash cement–soil was lower than that of plain cement–soil. The trend of the curves is the same. After 28 days, the strength of fly ash cement–soil has increased significantly, and a generalized hyperbolic function of composite soil strength increasing with time is proposed. Zhang Guirong et al. [21] used cement and FA to improve the fine sandy soil. In order to obtain the strength and permeability coefficient of the modified soil, a laboratory mechanical test was carried out. The results showed that increasing the FA admixture when the cement admixture was constant would greatly increase the compressive strength of the composite soil while decreasing its permeability coefficient; while increasing the cement admixture when the FA admixture was constant was beneficial to the increase in the compressive strength of the composite soil. Wang Minmin et al. [22] used dynamic triaxial tests to study the effects of polypropylene fiber and basalt fiber on the dynamic strength and dynamic elastic modulus of hydroclay. The results showed that the dynamic strength and dynamic elastic modulus of fibrous hydroclay were influenced by the surrounding pressure, fiber type, and fiber admixture; the dynamic strength and maximum dynamic elastic modulus of fiber content increased while the dynamic deformation decreased when the fiber content increased; the greater the surrounding pressure, the greater the maximum dynamic elastic modulus of fiber cement–soil. In order to study the influence of FA and polypropylene fiber on the mechanical properties, failure mode, and microstructure of cement–soil, Duan et al. [23] added 0%, 0.1%, 0.2%, 0.3%, and 0.4% of PP fiber and 0, 4, 8, and 12% of FA to cement–soil, and carried out the UCS test, split tensile strength test, and SEM test. Arıcı et al. [24] employed corn cob ash to modify polypropylene fiber-reinforced cement mortars, and they reported that a cement dosage of 22%, corn cob ash of 9%, and polypropylene fiber of 7% were effective via analysis of variance. The results showed that with the increase in PP fiber and FA content, the compressive strength and tensile strength of cement–soil first increased and then decreased, the strain curve of PP fiber–FA–cement–soil can be divided into four stages. The fiber limits the lateral deformation of the soil, thereby improving the peak strain and failure mode of the cement–soil.

Through the above literature review, it can be found that the existing studies are mostly based on single external admixture, such as the use of cement, FA, and so on to improve the silty soil. There are a few studies on the compound improvement of two or more external admixtures to improve the silty soil, and most of the common silty soil improvement is chemical improvement. There are a few results on the physical improvement of the silty soil using fiber, and there are few studies on the synergistic improvement of the silty soil combining physical and chemical effects. At the same time, it is found that there has been no in-depth study on the strength development tendency, characteristics of the stress–strain relationship, and analysis of microstructure and physical phase composition of PP fibers and FA modified cement silty soil. Based on this background, the study employed cement silty soil as the research object, incorporated with fly ash and polypropylene fiber to obtain a composite material with better deformation characteristics and higher strength, aiming to provide some basis for the engineering application of silty soil.

## 2. Experimental Program

### 2.1. Materials

All the soil samples used in this paper were taken from Kaifeng, Henan Province of China; the silty soil commonly exists in eastern Henan Province, and their physical and mechanical properties are shown in Table 1. According to the standard for the geotechnical testing method (GB/T 50123-2019) [25], the particle analysis test was conducted on the silty soil, and the particle distribution curve and particle analysis results of the soil samples are shown in Figure 1. The cement used in this investigation is P.O42.5 ordinary Portland cement, while low calcium fly ash and polypropylene fiber are applied as extra amendment materials.

### 2.2. Sample Preparation

In the modified soil samples prepared for this experiment, the cement admixture amounts were 4, 8, 12, and 16% of the dry soil weight. In order to study the effect of FA and PP fiber as external admixtures on the strength performance of the cement–soil and to determine the optimum admixture, four admixtures (0%, 0.15%, 0.3%, and 0.45% by weight of dry soil) of PP fiber and four admixtures (0%, 5%, 10%, and 15% by weight of dry soil) of FA were added to the soil; the test age was set to 7 d, 14 d, 28 d, and 60 d. The orthogonal test design was designed based on four factors and four levels; the orthogonal test design can obtain the optimal ratio with fewer and more comprehensive test combinations, and the orthogonal test design is shown in Table 2.

According to the Standard for “Test methods of Materials Stabilized with Inorganic Binder for Highway Engineering” (JTG E51-2009) [26], the preparation process of the test soil sample is shown in Figure 2, and the specific procedures are carried out in the following order: (1) Screening of the silty soil. The retrieved soil samples are air-dried and crushed, picked out of debris, and sieved so that they do not contain debris, weeds, etc. (2) According to the test plan, weigh the required quality with a high-precision electronic scale and add the weighed FA to the silt, cement, FA, PP fiber, and water, and stir evenly. (3) Add the weighed PP fiber to it and stir evenly (in order to prevent the fiber from clumping, it is added in a small amount several times, so that the fiber is evenly dispersed in the soil). (4) Then, add an appropriate amount of water (retaining 2–3% of the water content) and stir evenly, and then put the mixture into a plastic bag for 8 h. (5) Finally, 1 h before sample formation, add the weighed cement and reserved water to the FA–PP fiber–silt soil and fully stir, and store the prepared modified soil in a plastic bag. (6) Clean the test tube, let it dry for a period of time after wiping, apply a layer of lubricating oil on the inner wall and the surface of the cushion, weigh the mass of the mixture required for a single test piece, pour into the test tube at one time, adjust it so that the upper and lower cushion blocks are exposed to the same height, and then place it on the universal testing machine for compaction. After 2 h, put the test pieces on the demolding machine and number them. (7) Arrange the numbered soil samples neatly in the tray; then, cover the surface of all specimens with a layer of cling film and move them to the standard maintenance room (relative humidity above 95%, temperature 20 ± 2 ℃) for maintenance.

When carrying out the UCS test, the test can be carried out according to the above-mentioned test piece, and the sample was prepared by using a Φ50 × H50 mm cylindrical test mold. The SEM test requires cutting the specimen into larger pieces with a soil cutter, selecting a specimen with at least one flat section in the center of the specimen, and trimming it to a cube with a side length of about 1 cm using a soil trimmer; except for the freshly damaged surface, the remaining five surfaces can be cut flat. Before conducting the test, the specimen was glued to the small copper table using a carbon conductive adhesive, noting that the original damage section should be oriented upward as the observation surface, and the specimen fixed on the small copper table was put into the gold spray chamber and vacuumed before spraying gold. Figure 3 shows the specimen after gold spraying. When the specimen is maintained to the specified age, it is removed and cracked, and the specimen block at the center is put into the oven to dry at 30 °C for 12 h. Subsequently, the specimen was taken out and put into the grinding cup repeatedly, and the powder was poured onto a 0.075 mm sieve after grinding to a certain extent, and then, it was placed on a vibrating sieve machine for sieving treatment, with a sieve margin of no more than 10%. The sieved powder sample was sealed and stored in a catheter using a small spoon and numbered for XRD analysis, and the ground powder is shown in Figure 4.

### 2.3. Experimental Methods

A total of 16 sets of specimens were prepared in this experiment. For the UCS test, 6 parallel test specimens were made in each group to make the results more reliable. The WDW-20 type microcomputer-controlled electronic universal testing machine (The manufacturer of the equipment is Shanghai Precision Instrument Co., Ltd., and the place of production comes from Beijing, China.) was used for the UCS test, and the maximum load is 100 kN, which was loaded at a rate of 1 mm/min until the soil sample was damaged. When using the KYKY-EM6200 scanning electron microscope (The manufacturer of this equipment is Beijing Zhongke Science and Technology Co., Ltd., and the place of production comes from Beijing, China.) to conduct microscopic experiments on the sample soil, we first wiped the small copper table used to fix the sample with absolute ethanol in advance, cleaned the table to ensure cleanliness, and then used carbon conductive glue to stick the test block on the small copper table. Since the modified soil is a non-conductor, the observation quality and clarity of the electron microscope may be affected by the non-reflective electrons and the surface of the soil during observation. The electrical conductivity is improved by plating gold on its surface, and then, the changes within its microstructure are observed experimentally. The milled powder was used for XRD analysis using the test equipment as a D8ADVANCE (The manufacturer of the equipment is Bruker (Beijing) Technology Co., Ltd., and the place of production comes from Germany.) type diffraction analyzer.

## 3. Experimental Results

### 3.1. UCS Test

The UCS test results are analyzed by the range analysis method, which is intuitive, concise, and clear. The range analysis of PP fiber and FA cement-modified silt is shown in Table 3.

In Table 3, K_1j_ refers to the sum of the test indicators (UCS) corresponding to the j-th influencing factor at the first level, and j corresponds to the four influencing factors A (cement), B (fly ash), C (polypropylene fiber), and D (curing age). In the table, k_1j_ is the arithmetic mean value of the UCS corresponding to the j-th influencing factor at the first level. R_j_ is the range of the j-th factor. In the range analysis of the results of the orthogonal experiment, the R value reflects the importance of a certain factor to the test index; that is, the greater the range of a certain influencing factor, the greater the influence of the factor on the test results. In this test, the test index is the strength of the modified soil. Theoretically, the larger the value, the better. Therefore, the level corresponding to the largest K value among the factors is the optimal level. If only the strength is considered, the optimal solution is A_4_B_3_C_2_D_4_, which is a combination of 16% cement content, 10% FA content, 0.15% fiber content, and 60 d age.

The influence of changes in the levels of various factors on the UCS is shown in Figure 5.

It can be seen from Figure 5a that the UCS and the cement content change basically linearly, and the effect of cement on the strength improvement is very obvious. When the cement content is 16%, the UCS is increased by 113.4% compared with the cement content of 4%. Figure 5b shows that with the increase in the FA content, the strength of the modified silt tends to increase and then decrease, and it reaches the peak strength at 10% of FA content, and its UCS increases by 28.2% compared with 0% of FA content. When the amount of FA is small, the reaction between FA and cement hydration products is insufficient, and the increase in strength is not obvious. At this time, FA is mainly embedded in the cement–soil gap in the form of particles. The strength of the modified soil was greatly increased when the FA content was increased from 5% to 10%, at which time the active SiO_2_ and Al_2_O_3_ in FA react with the hydration product Ca(OH)_2_ to produce many cementitious products such as calcium silicate hydrate (C-S-H) and calcium aluminate hydrate (C-A-H), which greatly increase the strength of the modified soil. When the FA content exceeded 10%, the strength of the modified soil gradually decreases. This is because the increase in FA leads to the reduce of cement content and silty soil in the modified soil, and the corresponding skeletal structure formed between the cement and soil is reduced, thus leading to the decrease in strength. Figure 5c shows that with the increase in the PP fiber content, the strength of the modified soil first increases and then decreases, which is similar to the change trend of FA. When the PP fiber content is 0.15%, it can be seen that the UCS increases by 21% when compared with the content of 0%. When the initial fiber content is small, it can be seen from the figure that the strength has been greatly improved. This is because the fiber and the cement FA silt are nested together to form a spatial network structure, which improves the friction and occlusion force of the modified soil, and then improves the strength. After the fiber content exceeds 0.15%, the strength begins to decrease, which is because the excess fiber will appear in the soil in a larger bundle structure, leading to the local resistance to external load reducing and forming cracks, which eventually leads to the decrease in strength. It can be seen from Figure 5d that the longer the age, the higher the UCS of the modified soil. When the age is 60 d, the strength is increased by 152.2% compared with the UCS of 7 d. The strength of the modified soil increased faster in the first 28 days and slowed down after 28 days, but the strength always increased due to the time-dependent effects of cement hydration reaction and the reaction between hydration products and FA. The longer the time, the more adequate the reaction and the more the corresponding cementation products, while the transformation of these substances into the crystal structure greatly improves the strength of the modified soil. It can also be observed that the UCS reaches its peak when the cement content is 16%, the FA content is 10%, the PP fiber content is 0.15%, and the curing age is 60 d, which is consistent with the results of the range analysis.

In order to better understand the deformation characteristics and failure characteristics of the modified soil under uniaxial compression, the stress–strain curve of the PP fiber–FA–cement modified silt was studied, and based on this, guiding opinions for the actual project were put forward. Taking the 15th set of the orthogonal test as an example, we plot the measured stress–strain curve in Figure 6.

As shown in Figure 6, the stress–strain curve of PP fiber–FA–cement modified silt can be divided into three stages, namely: elastic deformation stage (stage I), plastic deformation stage (stage II), and strain-softening stage (stage III). In the elastic deformation stage, it can be clearly observed that as the stress continues to increase, the stress and strain of the modified soil change almost linearly. Due to the external load, the soil particles in the specimen were extruded but not damaged, and the modified soil began to harden. In the plastic deformation stage, before the stress reached the peak, the stress–strain curve began to fluctuate and rise, which is due to the uneven material distribution inside the modified soil, resulting in the unbalanced stress on the specimen and thus the fluctuation section; meanwhile, the strain changes obviously with the increase in stress until the peak stress appears. At this stage, the cement–soil and soil particle structure in the specimen began to break down, the particle gap gradually became smaller, and many small cracks appeared on the surface of the specimen. With the further increase in the stress, the width of the micro cracks increased. One or more main cracks and a large number of micro-cracks appear at this time. At the strain-softening stage, after the specimen reaches the peak stress, the stress decreases rapidly with the increase in strain and falls to the peak at about 30% of the peak stress, the stress–strain curve remains basically parallel to the *X*-axis, and the specimen starts to maintain a certain residual stress. At this stage, the cracks keep expanding outward and elongating until the specimen is completely damaged.

### 3.2. SEM Test

Modified soil is a relatively complex composite material, and the effect of external admixture on soil improvement is visually demonstrated by macroscopic tests, while its effect at the microscopic level is manifested by the change of material composition and structure of the products generated by a series of complex physicochemical reactions between soil–modifier and modifier–modifier. The microstructure essentially reveals the inner reason for the change of macroscopic mechanical properties. By observing the microstructure, the relationship between the composition of the material structure of modified soil at the microscopic level and the development of physical and mechanical properties at the macroscopic level is essentially elaborated, and based on this, the mechanism of action of various external admixtures for improving silty soil are analyzed. The SEM experiment was carried out using the KYKY-EM6200 electron microscope.

Figure 7 shows SEM images of plain cement–soil at different magnifications. As can be seen from the figure, SEM can observe the density of modified soil at low magnification. It can be seen from Figure 7a that there is a hydration product C-S-H gel attached to the surface of the soil particles at the observed failure surface to form a relatively dense skeleton structure, and a small amount of needle-like hydration products are generated, which indicates that a certain hydration reaction has occurred in the modified soil. However, larger soil particles can still be seen and the gaps between the particles are large, while the generated crystal products are short and fine, which cannot link large pore size pores, and more crystals developed in small pores. Therefore, when the cement content is low, the products of hydration have a limited effect on the strength enhancement of the modified soil, and the overall skeletal structure is loose. When the cement content is increased to 16%, Figure 7b shows that the framework is tightly connected and the pores between particles are less. Compared with the modified soil with low cement content, the structure is denser. The shadow in the lower left corner is caused by uneven surface, and the observation on the right side reveals that the modified soil is stacked by sheet-like products with relatively regular directions, the connection between soil particles and hydrates is tight, and the surface of the soil is loose, with fewer floating particles and a few small pores. The skeleton structure is the main load-bearing part of the cement–soil. The increase in the cement content makes the skeleton more dense and has a more obvious effect on the strength improvement. However, the excessive amount of cement not only increases the budget but also puts a load on the environment.

Figure 8 shows the SEM pictures of FA–cement–soil when the FA content is 5% and 10% respectively. It can be seen from Figure 8a that when the amount of FA is small, its main effect is to fill in the form of particles between the soil particles and the hydration product, leaving a part of the FA attached to the cement or soil. At this time, the overall structure of the modified soil is relatively dense, but most of the FA only plays the role of physical filling, and the cement products on the surface of soil particles are less, and there are more bare soil particles; that is, the effect of chemical modification of FA is weaker at a low mixing amount, and the effect of chemical modification of FA is weak. It can be found that the compounding of cement and FA can help improve the compactness of the modified soil, but when there is less FA, less cement can be generated, and the improvement effect is not obvious. It can be seen from Figure 8b that the cement increased, the soil particles are wrapped to form agglomerates, and the needle-like crystal products increased, which mostly appear in the gap between hydrate–hydrate and soil–hydrate. There are crystals in the pores, which play a strong bonding role. At this time, there are only a few small pores in the modified soil, and the overall structure is relatively compact. That is, the mixing of cement with an appropriate amount of FA promotes the generation of cement hydration products and the packaging of soil particles, thereby enhancing the compactness of the structure and effectively improving the strength of the soil.

Figure 9 shows the SEM images of the modified soil when the PP fiber content is 0.15% and 0.45%, respectively. In Figure 9a, the fiber content is small, the fibers are tightly embedded in the structure in a single bundle, and there are no obvious pores at the fiber–cement connection, and the mechanical occlusion of the two is strong. When the fiber is broken, the fiber surface shows longitudinal fracture cracks, which indicates that the fiber replaces the skeleton in the space structure to bear part of the external force, which can improve the damage form of the modified soil and reduce the brittleness of the cement–soil to a certain extent. In Figure 9b, the fiber content is too much, and then, a larger bundle-like structure is formed in the modified soil; that is, the fiber is not uniformly mixed, causing several fibers to be juxtaposed together. It can be clearly observed that at this time, the pores at the fiber–skeleton structure connection are large and numerous, the friction between the fiber and the hydration product is reduced, the large bundle structure causes slippage between the two, and the soil structure tends to be unstable.

Figure 10 shows the SEM pictures of fiber FA cement–soil at different ages when the cement content is 16%. According to the microscopic picture in Figure 10a, it can be seen that the cement hydration products in the modified soil after curing for 14 days are dense, most of which are distributed on the surface of the soil particles in the form of needles and rods, and the small amount of flaky products makes it difficult to see the large soil particles. Meanwhile, the bulging of spherical particles indicates that the FA has strong activity, reflecting the high degree of chemical reaction inside the soil. However, the surface flatness of the modified soil is low, and the cementation products are unevenly attached to the surface of the soil particles, with more small pores, and even longer crevices can be seen. This shows that the increase in the curing age promotes the chemical reaction of the cement, FA, and soil in the modified soil, and it generates a large amount of hydration products, which improves the strength of the modified soil and increases its compactness to a certain extent, but the flatness of the entire soil structure is low. Figure 10b shows the modified soil at the curing age of 28 days. Comparing with Figure 10a, it is obvious that the surface of the soil has become smooth, the cemented product is covered with soil particles, and the small gaps are filled with needle-like products, which reduces the porosity of the soil. This shows that when the age increases from 14 to 28 days, the microscopic framework structure continues to grow, the internal structure of the modified soil is tighter, and the cement product is limited to connect the soil particles, which are better developed in both large and small pores, and thus, they improve the ability of the soil to resist external loads. This also verifies the macroscopic law of the change of the mechanical properties of the modified soil; that is, the strength of the modified soil increases rapidly during this period.

### 3.3. XRD Test and Analysis

The test equipment is a D8 ADVANCE diffraction analyzer. Based on the orthogonal test, the XRD test was performed on some fiber FA–cement–soil, and the diffraction patterns are shown in Figure 11, Figure 12, Figure 13 and Figure 14. The physical phase analysis of modified soil samples with different mixing ratios found that no matter how much modifier is added or the age, the diffraction patterns of the fiber FA cement modified silt are basically the same; that is, the material composition of the modified soils did not change, mainly SiO_2_, AFt, C-S-H. However, the intensity of the diffraction peaks of the corresponding phases in the XRD patterns of the specimens with different fit ratios differed. The analysis shows that the diffraction peak of SiO_2_ in the modified soil sample is the highest, indicating that it occupies the largest proportion. This is because the soil content in the sample is the highest (Figure 11), which means that the content of SiO_2_ is the highest, thus causing the high diffraction peak of SiO_2_. From the XRD pattern of the plain cement–soil in Figure 12, it can be observed that obvious ettringite (AFt) diffraction peaks and C-S-H diffraction peaks have appeared when the age is only 7 days, indicating that the hydration reaction of cement occurs in a relatively short period of time, and certain hydration products are formed. The increase in cement content and curing age makes the diffraction peaks of C-S-H increase significantly, while the diffraction peaks of AFt increase slightly; that is, more hydration products are generated inside the soil. According to the previous analysis of macroscopic mechanical laws, it can be concluded that cement and age play a positive role in the increase in strength of modified soil.

Figure 13 shows the XRD diffraction patterns of cement–soil mixed with FA and without FA at different ages when the cement content was 12%. In the figure, in the cement–soil cured for 60 days, the diffraction peaks of AFt and C-S-H are higher; while in the cement–soil mixed with FA and cured for 14 days, due to the short curing time, the diffraction peaks of AFt and C-S-H are lower, which indicates that the age promotes the generation of hydration products far more than FA. Comparing the plain cement–soil in Figure 12, it can be found that the diffraction peaks of the hydration products of the modified soil with cement compounded with FA (14 d) are between those of the modified soil with 4% cement (7 d) and those of the modified soil with 8% cement (28 d), which indicates that the addition of FA can promote the formation of chemical products inside the soil. In terms of the diffraction peaks of hydration products, it can be found that FA can increase the internal chemical reaction activity of the modified soil, and it has a certain effect on promoting the reaction, but it is not as large as the effect of age.

From Figure 14, the XRD diffraction profiles of cement–soil with different FA contents at different curing ages when the cement content is 16% are shown. Comparing the four sets of XRD curves, it can be found that when the cement content is constant, the diffraction peaks of AFt and C-S-H in the modified soil cured for 60 days are the highest. The longer the age, the higher the diffraction peaks of hydration products, but the diffraction peaks increase and then decrease with the increase in FA. That is, the diffraction peak of the hydrated product is the highest when the FA content is 10%, which is consistent with the macroscopic mechanical properties. Meanwhile, comparing the specimens with different maintenance periods, it can be found that the generation of hydration products in the middle and late stages of the modified soil is more relative to the early stage; that is, the increase in soil strength is more obvious at longer ages.

## 4. Curing Mechanism of PP Fiber–FA–Cement–Soil

The strength of PP fiber–FA–cement-modified soil is a result of the multiple complex physical and chemical reactions carried out, and the main source of its strength increase is the cement skeletal structure formed by the hydrolytic hydration reaction of cement, and the chemical reactions within the composite soil are carried out more fully with the extension of time. The role of FA, as a more active granular material, in the strength growth of modified soils is mainly physical in the form of filling and chemical in the role of hydration. PP fibers are polymers with stable chemical properties, which mainly play a physical reinforcing role.

### 4.1. Curing Mechanism of Cement–Soil

#### 4.1.1. Hydrolysis Hydration Reaction of Cement

From the XRD diffraction analysis of the cement modified silt, it can be seen that the main products of the chemical reaction in the cement–soil are C-S-H and AFt. Generally, the skeleton structure of cement–soil is represented by the hydration product of cement, cementing soil particles, and filling pores in the form of composition. In the process of forming the strength of cement–soil, the first effect is the hydration and hydrolysis of cement. This process is very complicated because there are many types of mineral components in ordinary Portland cement, mainly tricalcium silicate (C_3_A), dicalcium silicate (C_2_A), tricalcium aluminate, tetracalcium aluminate, calcium sulfate, etc. These components will undergo many chemical reactions when they meet with water, resulting in a large amount of gel and crystalline substances.

#### 4.1.2. The Interaction between Silt Particles and Cement Hydrate

Soil is a mixed system that contains soil particles of different sizes and different components, which create a connection and make the soil have a certain strength [27]. The silt particles have certain adsorption properties, and when cement is added, they rapidly undergo hydration and hydrolysis reactions, resulting in the production of large amounts of hydrated calcium silicate and Ca(OH)_2_. Some of these products continue to solidify and harden, forming needle-shaped AFt and C-S-H gels, which form a larger skeletal structure inside the soil and thus increase the overall resistance of the soil to external loads. The remaining part of the hydration products will continue to react chemically with the active substances SiO_2_ and Al_2_O_3_ in the soil particles to produce the corresponding gelling substances with a certain strength, which closely connect the whole soil structure together and strengthen the denseness of the space structure. The overall effects are mainly ion exchange and agglomeration, volcanic ash reaction, crystallization, and carbonation.

### 4.2. Curing Mechanism of FA on Cement–Soil

#### 4.2.1. The Physical Filling Effect of FA

The particles of FA are fine, and the specific surface area is large. After the FA and cement–soil are mixed evenly, the more active FA will react chemically with the hydration products and so on. The other part of inert FA cannot undergo chemical reaction, but due to the small particle size, it can be directly embedded in the small pores between the cement–soil structure, filling the small pores completely, making the whole soil space structure more compact, and improving the strength of the FA–cement–soil. It can be observed from the mechanical test law in the previous paper that there is a peak in the strength of the composite soil with the increase in FA content. In this paper, the optimum FA content is 10%. When the content of FA is low, the volcanic ash reaction products are less, and the enhancement of strength is not obvious. At this time, FA mainly plays a filling role.

#### 4.2.2. Hydration Effect of FA

When FA is mixed into cement–soil, a series of chemical reactions will occur, which is called the hydration effect of FA. The main active substances in FA are SiO_2_ and Al_2_O_3_, and these active substances will react with Ca(OH)_2_ in the hydration products in a relatively strong volcanic ash reaction as well as the generated cementitious substances such as C-S-H and C-A-H. The condensate will connect the soil particles closely, reduce the voids inside the soil, and increase the strength of the soil. In this paper, the maximum admixture of cement is 16%, and the amount of Ca(OH)_2_ generated by cement due to hydration is limited, so the volcanic ash reaction will stop after Ca(OH)_2_ is consumed. In addition, XRD diffraction analysis of FA shows that anhydrite (CaSO_4_) is contained in FA. The C_3_A in cement will produce tricalcium aluminate hydrate when it meets with water, and it will chemically react with gypsum in FA to produce high-sulfur calcium sulfoaluminate hydrate crystals, namely needle-shaped ettringite [28]. After gypsum participates in the reaction, if the quantity is insufficient, low-sulfur calcium sulfoaluminate hydrate crystals will be formed. The reaction equations of the two are shown in the following Equations (1) and (2). These crystalline substances are distributed in the pores inside the cement–soil and play a role in connection. At the same time, the products of the pozzolanic reaction of FA will also chemically react with gypsum to form crystalline substances, and calcium aluminate hydrate will condense with each other and then form a network and flocculent structure on the surface of the soil particles, and all these reactions have a positive effect on the strength of the specimen.
(1)$$3\mathrm{Cao}\cdot {\mathrm{Al}}_{2}O_{3}\cdot 6H_{2}{O+3(\mathrm{CaSO}}_{4}\cdot 2H_{2}O)+20H_{2}O\to 3\mathrm{Cao}\cdot {\mathrm{Al}}_{2}O_{3}\cdot {3\mathrm{CaSO}}_{4}\cdot 32H_{2}O$$
(2)$$3\mathrm{Cao}\cdot {\mathrm{Al}}_{2}O_{3}\cdot 6H_{2}{O+3(\mathrm{CaSO}}_{4}\cdot 2H_{2}O)\to 3\mathrm{Cao}\cdot {\mathrm{Al}}_{2}O_{3}\cdot {3\mathrm{CaSO}}_{4}\cdot 12H_{2}O$$

The hydrolysis hydration reaction of cement is synchronized with the chemical reaction of FA. The hydration products of cement will have a pozzolanic reaction and hydration reaction with the active substances in FA, but the amount of Ca(OH)_2_ in the hydration products of cement and gypsum in FA is limited, so when a series of chemical reactions of FA are finished, the hydration reaction of cement is mainly at work inside the composite soil, and this process will generate more gel products to fill the soil particle gap, so that the internal structure of modified soil is more solid, significantly improving the ability of the soil to resist external forces and deformation. Due to the more stable structure of the glass beads of FA, the hydration reaction of FA is slower compared to the hydration of cement due to this limitation. The strength of cement–soil mainly comes from the products generated by a series of physical and chemical reactions between cement and soil particles in contact with water and the skeletal structure formed by them. Adding more FA to the cement–soil means that there is less material for the cement to react with, so the hydrolysis hydration products of the cement are reduced and the corresponding skeletal structure is less, so too much FA admixture will instead reduce the strength of the soil and its ability to resist deformation.

### 4.3. Curing Mechanism of PP Fiber on Cement–Soil

A certain percentage of fiber is mixed into the cement–soil, which is fiber-reinforced soil. The tensile strength of the fiber is high, and it is evenly mixed into the cement–soil so that the fibers are randomly distributed in all directions of the soil, and forces occur at the connection surface between the soil and the reinforcement, which in turn limits the lateral and longitudinal displacement of the soil, so that the fiber-reinforced soil can also be considered as a homogeneous composite building material [29]. The reinforcement mechanism of fiber-reinforced soil is different from that of conventional-reinforced soil. Ordinary reinforced soil consists of laying flaky or strip-shaped reinforcements in the same direction in the soil. When the soil is subjected to external forces, the friction between the soil reinforcements restrains the deformation of the soil, and the reinforced materials are subjected to tensile stress. Generally, the failure form of reinforced soil is tensile damage or adhesion damage, and the difference between the two is that the main influencing factors at the time of damage are not the same [30]. The distribution of fibers in soil can be regarded as uniform, multi-directional random, and mesh interweaving, so the reinforcement and strengthening mechanism of fibers is more complicated. PP fiber is a synthetic fiber with a low modulus of elasticity and stable chemical properties, and its reinforcement mechanism for soil is physical reinforcement. Current research shows that the curing of soil by fibers comes not only from the frictional effect between fibers and soil particles but also from the influence of the spatial mesh structure formed between fibers and fibers.

#### 4.3.1. Friction Reinforcement Effect

When PP fibers are mixed into the cement–soil, the specimen is formed with many fiber–soil particle connection surfaces, and under the action of external stresses, the soil reinforcement contact surfaces will exert certain bonding and frictional resistance to resist external forces, thus improving the force properties and deformation characteristics of the soil. The principle of friction reinforcement of a certain micro-segment of composite soil under the action of PP fiber is selected to analyze the principle of friction reinforcement [31], as shown in Figure 15.

Supposing that the length of the micro-segment is *dl*, the tensile forces *T*_1_ and *T*_2_ are applied at the cross-sections of both ends of the PP fiber, and the cross-sectional area is S. The normal pressure on the outer surface of the PP fiber is *N*, and the coefficient of friction between the fiber and the soil is *µ*. The quality of fibers and soil particles is negligible. When the soil is subjected to an external load, the external strain will cause the fiber on the micro-segment to generate tensile stress *dT*; then, *dT* = *T*_1_ − *T*_2_, assuming that the friction generated on the micro-segment is *dF*. Then, there is:(3)dF=μ·2N·S·dl.

When the total friction between the PP fiber and the soil is greater than the tensile stress caused by the external load, that is, *dF* > *dT*, there will be no relative slippage between the soil particles and the fiber on the micro-section, that is, the frictional resistance between soil bars overcomes the internal stress generated by external load, so the cement–soil within this segment can maintain its original structure and limit the development of internal cracks. Secondly, inside the modified soil, the fibers are randomly and evenly distributed and staggered and connected, so a spatial fiber–fiber interwoven network structure is formed, which can restrain the deformation of the soil.

When PP fiber is mixed into FA–cement–soil, the cement and FA chemical reaction generate gelling substances, and crystals cover the outer surface of the fiber and soil particles, increasing the bonding stress and frictional resistance between the fiber and the soil particles. When subjected to external forces, micro-cracks will appear at the weak points of the internal structure of the composite soil. At this time, the interface force between the fiber and the cement–soil hinders the further development of the cracks. With the further increase in the external load, the weak points inside the cement–soil are damaged. The PP fiber transfers the external load to the fiber–soil connection surface due to its good tensile properties, and through the mechanical interaction of the fiber and the soil particles and the linkage between the fibers, the external force is shared, which limits the development of cracks and transforms the brittle damage of the cement–soil to plastic damage at the same time. It can be seen from the above analysis that when the fiber content is small, the fiber distribution is relatively dispersed and cannot produce an interweaving effect. Therefore, the improvement of the mechanical properties of the cement–soil by the fiber is not very significant. After the fiber content is appropriately increased, the interweaving effect will occur. At this time, the friction reinforcement, bending mechanism, and interweaving mechanism between the fiber–soil particles and the fiber–fiber work simultaneously, so that the cement–soil resistance to external force and deformation can reach a better state. However, when the amount of fiber is too much, the internal bundle structure of cement–soil is more, so that the bonding force and friction between fiber–soil are reduced, and the linkage between fiber–fiber is not strong, so the mechanics of cement–soil performance and toughness are reduced.

#### 4.3.2. Lateral Restraint Effect

The lateral restraint effect means that the fibers are uniformly dispersed in the soil, which will produce a lateral restraint on the lateral deformation of the soil, thus increasing the cohesive force inside the cement–soil. When the cement–soil is subjected to external load, the friction between PP fibers and soil particles will prevent the relative sliding of the two, which is equivalent to the PP fibers imposing a constraint parallel to the displacement direction in the soil, making the internal force of the cement–soil smaller than the external force and improving the strength and deformation characteristics of the soil [32].

When the unreinforced cement–soil is subjected to vertical stress $\sigma_{1}$, the soil will be compressed, and with the increase in $\sigma_{1}$, the soil will undergo transverse and longitudinal deformation until it is completely destroyed, and its basic stress state diagram is shown in Figure 16a. While the reinforced cement–soil is under the action of $\sigma_{1}$, its lateral deformation and longitudinal deformation after the damage will be significantly reduced, and the stress state of PP fiber-reinforced cement–soil is shown in Figure 16b.

When acts on the fiber-reinforced cement–soil, the modified soil will produce vertical compression and lateral expansion deformation, but the elastic modulus of the fiber is higher than that of the cement–soil, so greater deformation will occur in the cement–soil. According to the principle of deformation coordination, the fiber will produce a lateral constraint on the soil to limit its lateral deformation. The constraint stress is $\sigma_{R}$, which is equivalent to applying a lateral pressure $∆\sigma_{3}$ to the soil. The following stress molar circle is used to show the reinforcing effect of fibers more visually.

As shown in Figure 17, according to the state of unreinforced cement–soil under unconfined compression, the limit stress molar circle I can be obtained. It can be seen from the above that the fibers in the fiber-reinforced cement–soil will produce a lateral pressure $∆\sigma_{3}$ on the soil, which can be equivalently regarded as the unreinforced cement–soil simultaneously subjected to vertical pressure and lateral confining pressure. Based on the triaxial principle, the ultimate stress molar circle II of the “unreinforced cement–soil” at this time is obtained. Therefore, the failure envelope of the unreinforced cement–soil is the straight line AB tangent to the stress molar circle I and II, the cohesive force c is the intersection of AB and the *Y*-axis, and the internal friction angle *φ* is the angle between AB and the *X*-axis.

The fibers are arranged in a staggered arrangement and connected to each other when mixed into the soil, thereby forming a spatial network structure to wrap the soil particles or enhancing the bonding stress between the soil particles through the “bending mechanism”, so the addition of fibers has a greater impact on the cohesion of the soil, but it does not affect the arrangement and surface roughness of the soil particles. The internal friction angle of reinforced cement–soil is almost not influenced by the fiber admixture, and the difference with the internal friction angle of unreinforced cement–soil is very small. Accordingly, if the failure envelope of the fiber-reinforced soil–cement circle Ⅲ is A′B′, then A′B′ and AB are almost parallel and tangent to circle Ⅲ. Figure 17 shows that the cohesive force of fiber-reinforced hydroclay increases significantly compared with that of unreinforced hydroclay, while the angle of internal friction does not change significantly. Therefore, the improvement of the strength of the fiber on the cement–soil is characterized by the increase in the cohesive force, and the essence is that the lateral restraint of the fiber on the cement–soil body limits its deformation, while at the same time, the trend of the deformation and damage of the cement–soil gradually changes from brittle to plastic.

## 5. Conclusions

In this paper, the mechanical properties, curing mechanism, and microstructure of FA and PP fiber synergistically modified cement–silty soil were investigated by conducting UCS, SEM, and XRD experiments. The main conclusions are as follows.(1)With the increase in FA and fiber, the strength first increases and then decreases. There is an optimal content of the two. Among them, FA is 10% and fiber is 0.15%. At the same time, combined with the range analysis method to target the strength, the best mix ratio is determined as follows: the cement content is 16%, the FA content is 10%, the PP fiber content is 0.15%, and the curing age is 60 d. At the same time, it is found that the stress and strain development of the modified soil has three stages: namely, the elastic stage, the plastic deformation stage, and the strain-softening stage, and the specimens in each stage had corresponding deformation characteristics.(2)It can be seen from SEM images that a large amount of cementitious substances and needle-like and layered hydrates are produced in the PP fiber–FA–cement–soil. These products cover the surface of the soil particles or fill the pores between the soil particles, forming a skeleton structure and fiber–soil spatial network structure, enhancing the integrity of the modified soil structure; PP fiber is embedded in the soil particles, and the connection between the two is wrapped in many hydration products, thereby increasing the friction and mechanical bite force between the fibers and the soil–cement particles.(3)The mineral composition of the modified soil with different mixing ratios is basically the same, mainly SiO_2_, C-S-H, and AFt, but the diffraction peak intensity of each mineral was different. The samples with large cement content and long curing age have a higher diffraction peak intensity of calcium silicate hydrate and ettringite. The amount of FA has little effect on the intensity of diffraction peaks of mineral components in the modified soil. All in all, the diffraction peak intensity of the mineral composition in the modified soil is greatly affected by the cement and age, and the effect of FA is weak.(4)The improvement effect of cement on silt is mainly reflected in the hydrolysis and hydration reaction of cement and the chemical reaction between soil particles and hydration products. The chemical strengthening effect of the two makes the strength of cement–soil greater. The curing effect of FA on cement–soil is reflected in its physical filling and hydration effects, while fiber-reinforced cement–soil mainly has friction reinforcement and lateral restraint effects.

## Figures and Tables

**Figure 1 materials-14-05441-f001:**
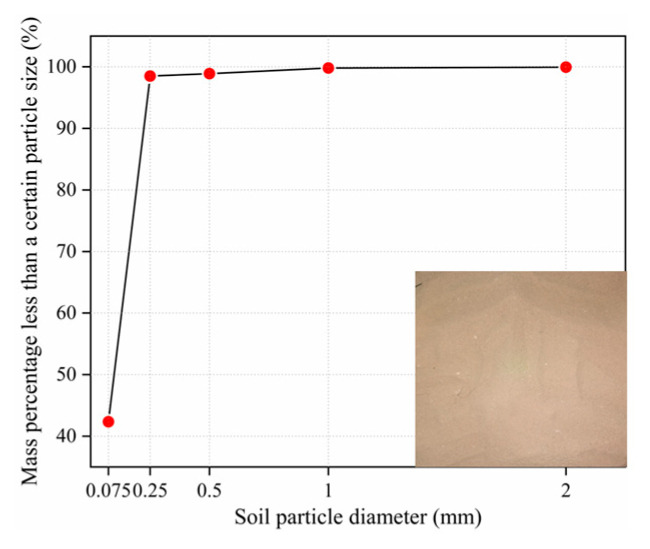
Particle gradation curves of soil samples.

**Figure 2 materials-14-05441-f002:**
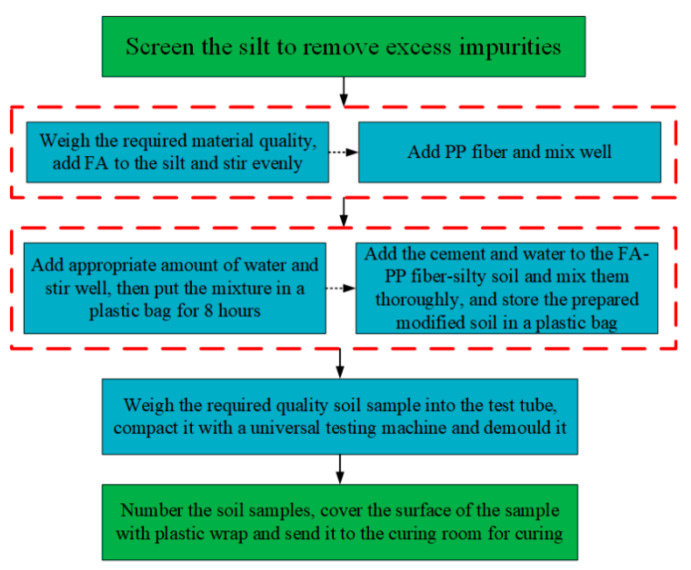
Test soil sample preparation process.

**Figure 3 materials-14-05441-f003:**
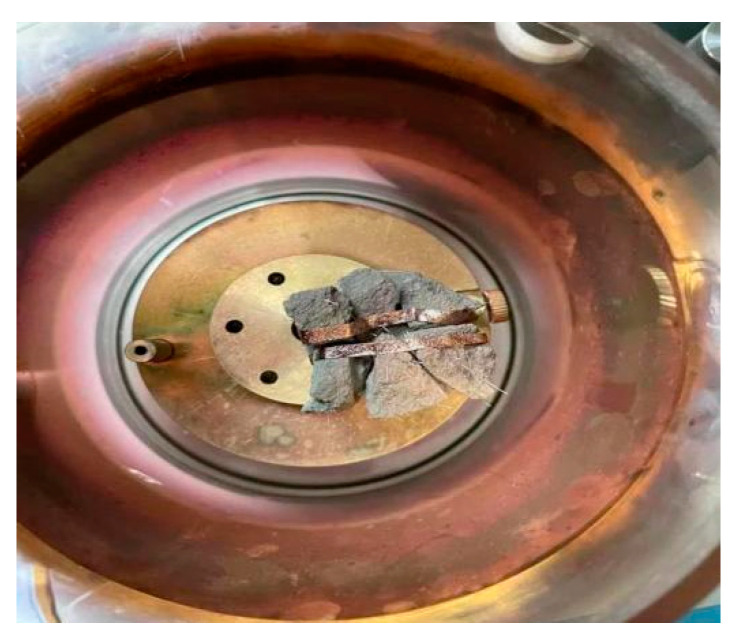
Gold-sprayed sample.

**Figure 4 materials-14-05441-f004:**
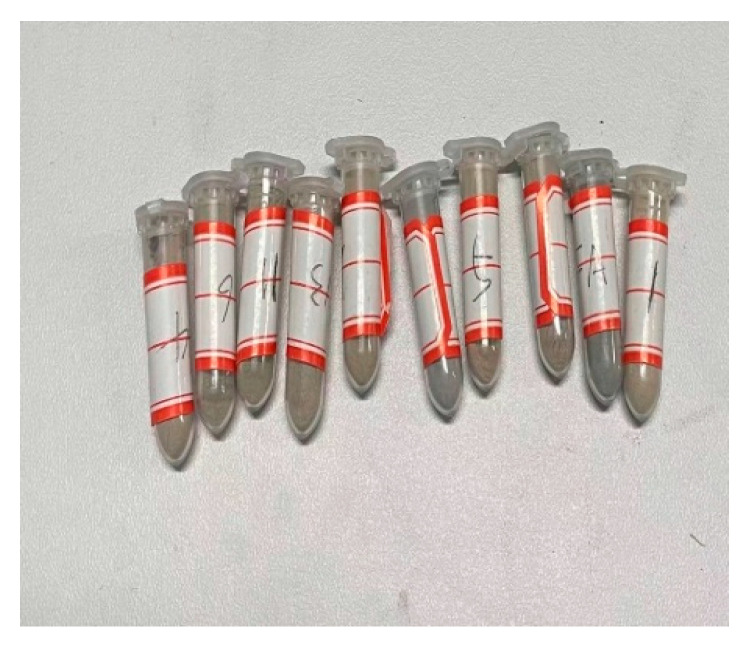
XRD test powder.

**Figure 5 materials-14-05441-f005:**
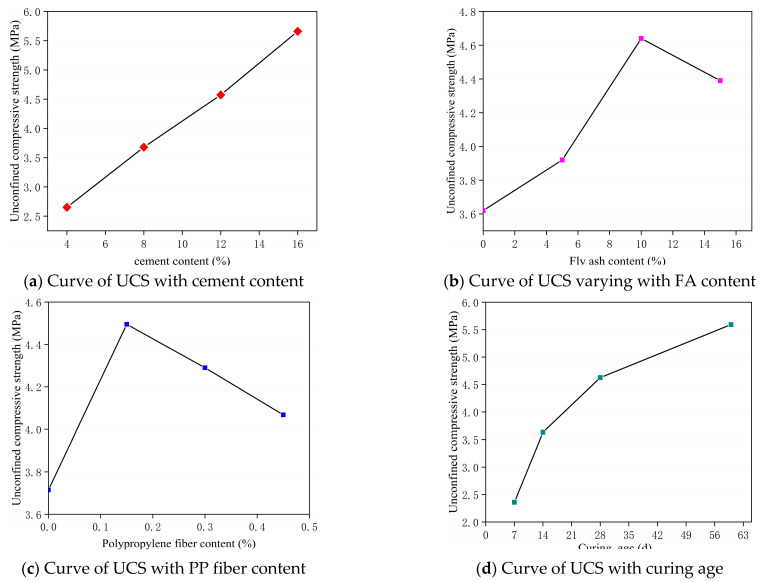
The impact of UCS with the level of various influencing factors.

**Figure 6 materials-14-05441-f006:**
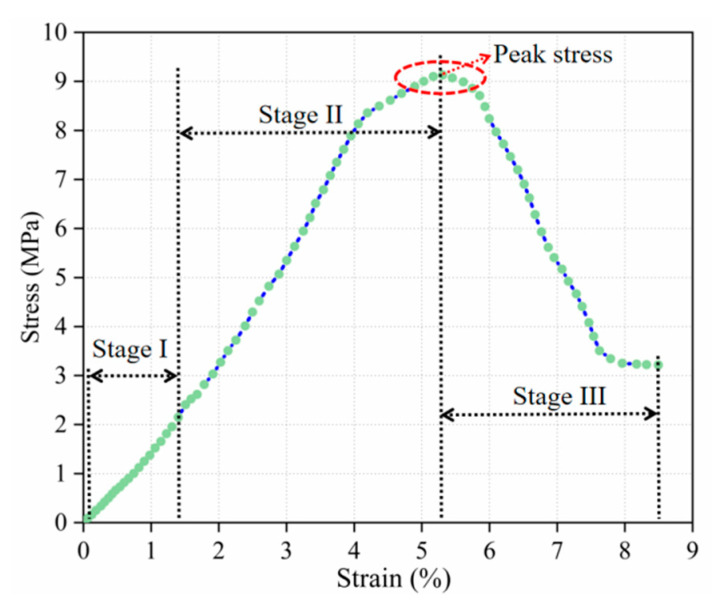
Measured stress–strain curve.

**Figure 7 materials-14-05441-f007:**
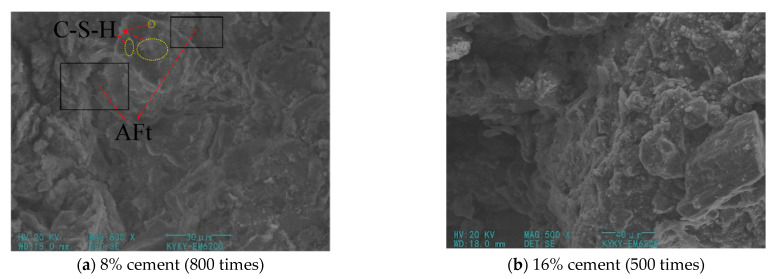
SEM picture of plain cement–soil.

**Figure 8 materials-14-05441-f008:**
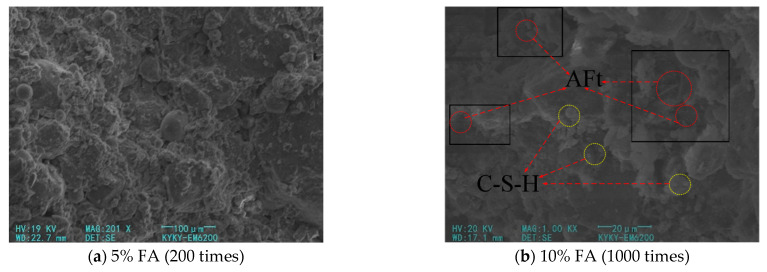
SEM pictures of modified soil with different FA content.

**Figure 9 materials-14-05441-f009:**
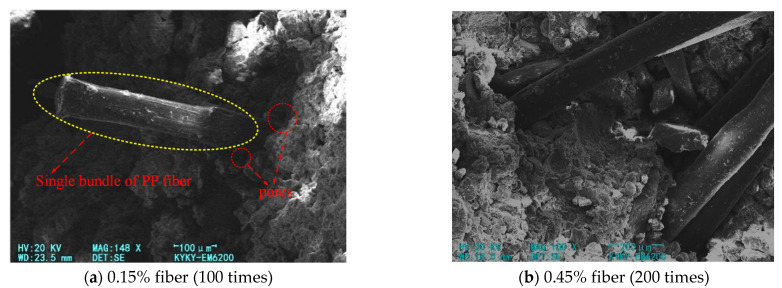
SEM picture of fiber FA cement–soil.

**Figure 10 materials-14-05441-f010:**
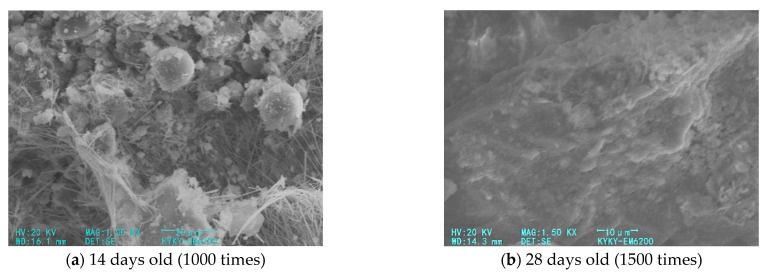
SEM pictures of fiber FA cement–soil at different ages.

**Figure 11 materials-14-05441-f011:**
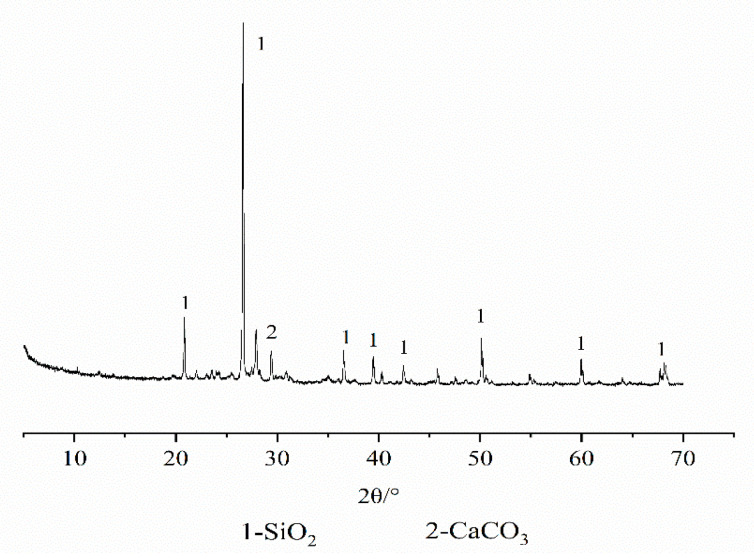
XRD diffraction pattern of silt.

**Figure 12 materials-14-05441-f012:**
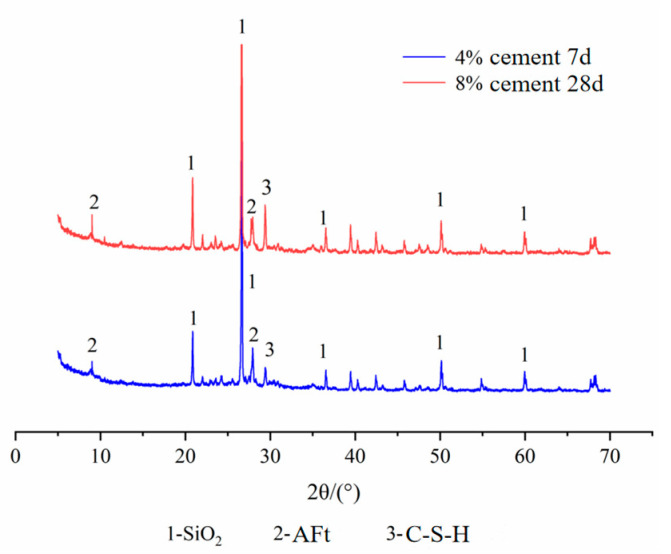
XRD diffraction pattern of plain cement–soil.

**Figure 13 materials-14-05441-f013:**
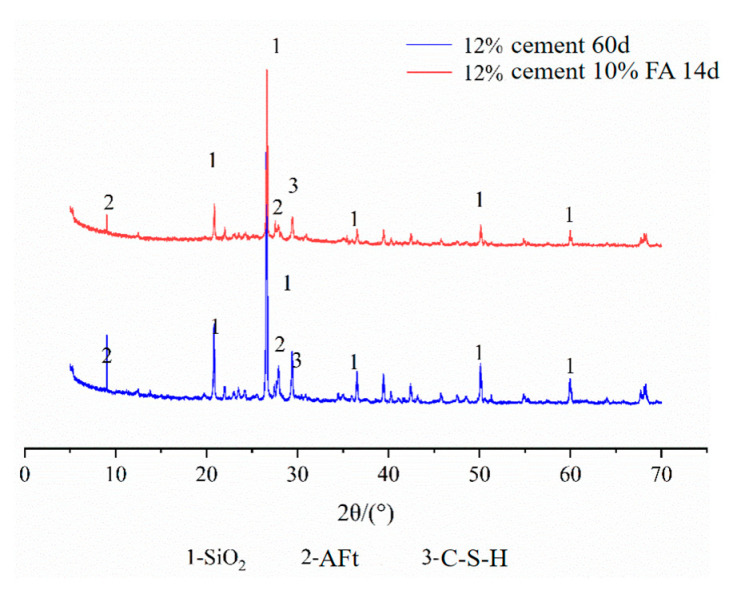
XRD diffraction pattern of FA cement–soil.

**Figure 14 materials-14-05441-f014:**
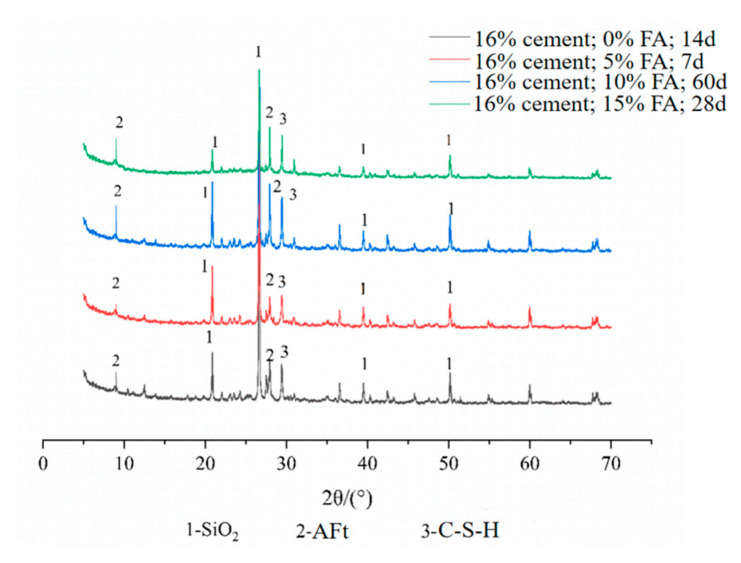
XRD diffraction pattern of cement–FA-modified soil.

**Figure 15 materials-14-05441-f015:**
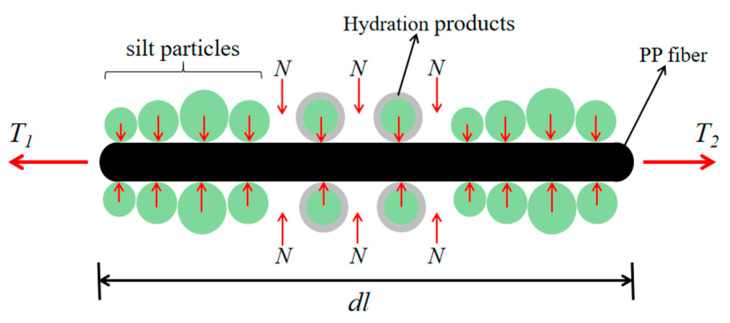
Schematic diagram of the micro-segment of friction reinforcement.

**Figure 16 materials-14-05441-f016:**
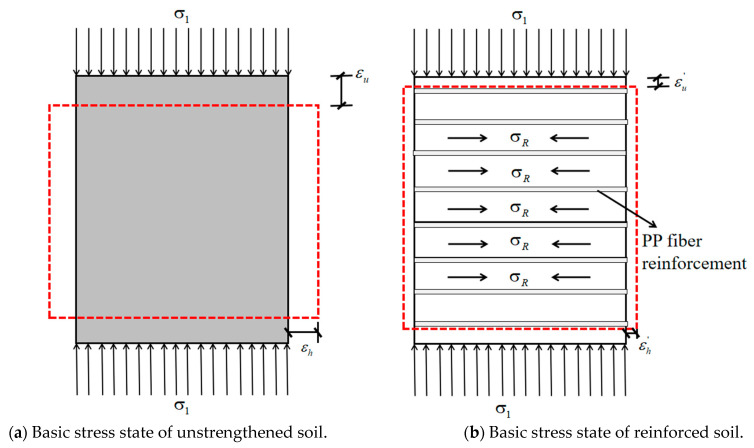
Basic stress state of cement–soil before and after reinforcement.

**Figure 17 materials-14-05441-f017:**
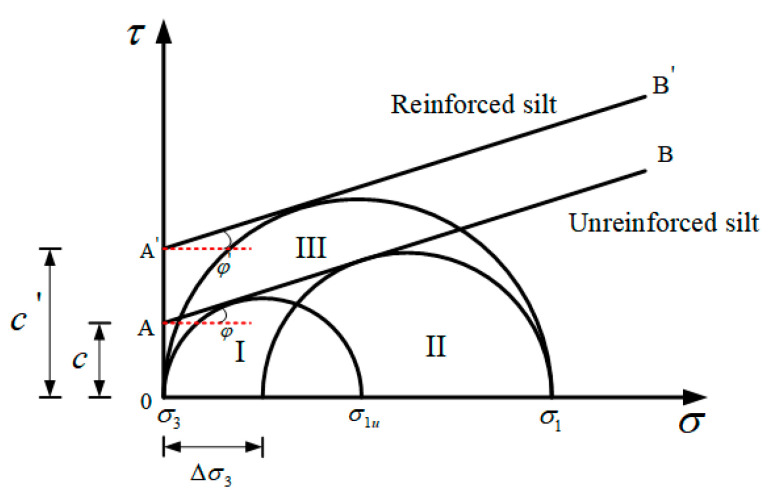
Stress Mohr circle of fiber-reinforced soil.

**Table 1 materials-14-05441-t001:** Physical properties of silty soil.

Parameters of Silty	Numerical Value
Liquid Limit (%)	23.16
Plastic Limit (%)	15.35
Plasticity Index	7.81
Maximum Dry Density (g·cm^−^^3^)	1.784
Optimum Water Content (%)	11.5

**Table 2 materials-14-05441-t002:** Orthogonal test table.

Experimental Group	Experimental Factors			
	Cement Content (%)	FA Content (%)	PP Fiber Content (%)	Curing Age (d)
1	4	0	0	7
2	4	5	0.15	14
3	4	10	0.30	28
4	4	15	0.45	60
5	8	0	0.15	28
6	8	5	0	60
7	8	10	0.45	7
8	8	15	0.30	14
9	12	0	0.30	60
10	12	5	0.45	28
11	12	10	0	14
12	12	15	0.15	7
13	16	0	0.45	14
14	16	5	0.30	7
15	16	10	0.15	60
16	16	15	0	28

**Table 3 materials-14-05441-t003:** Range analysis of orthogonal test results of fiber FA cement-modified silt.

No.	C (%)	FA (%)	PPF (%)	A (%)	UCS (MPa)
1	4.00	0.00	0.00	7.00	0.82
2	4.00	5.00	0.15	14.00	2.46
3	4.00	10.00	0.30	28.00	3.25
4	4.00	15.00	0.45	60.00	4.08
5	8.00	0.00	0.15	28.00	3.44
6	8.00	5.00	0.00	60.00	4.31
7	8.00	10.00	0.45	7.00	2.51
8	8.00	15.00	0.30	14.00	4.46
9	12.00	0.00	0.30	60.00	6.20
10	12.00	5.00	0.45	28.00	5.66
11	12.00	10.00	0.00	14.00	3.58
12	12.00	15.00	0.15	7.00	2.86
13	16.00	0.00	0.45	14.00	4.02
14	16.00	5.00	0.30	7.00	3.25
15	16.00	10.00	0.15	60.00	9.22
16	16.00	15.00	0.00	28.00	6.15
K_1j_	10.61	14.48	14.86	9.44	
K_2j_	14.72	15.68	17.98	14.52	
K_3j_	18.30	18.56	17.16	18.50	
K_4j_	22.64	17.55	16.27	23.81	
*n*	4.00	4.00	4.00	4.00	
k_1j_	2.6525	3.62	3.715	2.36	
k_2j_	3.68	3.92	4.495	3.63	
k_3j_	4.575	4.64	4.29	4.625	
k_4j_	5.66	4.39	4.0675	5.9525	
R_j_	3.0075	1.02	0.575	3.5925	

C indicates the amount of cement; FA indicates the amount of fly ash; PPF indicates the amount of polypropylene fiber; A indicates curing age.

## Data Availability

Not applicable.

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
