# Peer review of "Mechanical Properties, Curing Mechanism, and Microscopic Experimental Study of Polypropylene Fiber Coordinated Fly Ash Modified Cement–Silty Soil"

_materials, 2021, doi:10.3390/ma14185441_

Round 1

Reviewer 1 Report

The manuscript "Mechanical properties, curing mechanism and microscopic experimental study of polypropylene fiber coordinated fly ash modified cement-silty soil" is very relevant and adheres to the theme of this research, but some corrections should be made by readers:

a) In the introduction, there are some gaps in the literature that the authors should insert, thinking about it, some works with a related theme should be read and added, such as: 10.1520/JTE20210054; 10.1016/j.cscm.2021.e00652; 10.4028/www.scientific.net/MSF.798-799.548.

b) In the final section of the introduction, the objectives must be highlighted by the authors, this is not clear at the end of this section;

c) In the materials and methods section, there is a sequence of figures and tables, this should be avoided, and intermediate texts should be inserted;

d) There are several images of equipment (and others), this is not necessary, please remove;

e) The quality of most micrographs is not adequate, can authors improve this quality?

f) The conclusion must be revised, think about reducing it and making it more objective, besides putting the conclusions in points can be interesting for readers.

Author Response

1. In the introduction, there are some gaps in the literature that the authors should insert, thinking about it, some works with a related theme should be read and added, such as: 10.1520/JTE20210054; 10.1016/j.cscm.2021.e00652; 10.4028/www.scientific.net/MSF.798-799.548.

Response: In light of your recommendation, we have deeply reviewed the above three articles, among them, the second one namely ‘An Analysis of the Engineering Properties of Mortars Containing Corn CobAsh and Polypropylene Fiber Using the Taguchi and Taguchi-Based Grey Relational Analysis Methods’ is closely related to our study, therefore we cited it as one of the evidence to illustrate the modified effect of fiber (line 139-142).

2. In the final section of the introduction, the objectives must be highlighted by the authors, this is not clear at the end of this section

Response: We have modified the final paragraph to emphasize the main study contents (line 156-159).

3. In the materials and methods section, there is a sequence of figures and tables, this should be avoided, and intermediate texts should be inserted;

Response: In this section, the unnecessary tables and figures were deleted to highlight the main physical and mechanical properties of silty soil, i.e. Table 2-5 and Fig 2-3.

4. There are several images of equipment (and others), this is not necessary, please remove

Response: These figures about equipment were deleted. 

5. The quality of most micrographs is not adequate, can authors improve this quality?

Response: We are really sorry for this, the main shortage of our equipment is the slightly worse imaging quality in high-expansion. Due to the tests were performed a year ago, it is not possible to improve the quality of these figures.

6. The conclusion must be revised, think about reducing it and making it more objective, besides putting the conclusions in points can be interesting for readers.

Response: We have revised the conclusion point by point.

Reviewer 2 Report

Congratulations to the authors of a very interesting research work. Determining the impact of the amount of cement, ash and polypropylene fiber on the improvement of soil properties is an important area that can be directly applied in road and hydrotechnical construction.

In my opinion, the use of polypropylene fiber is economically unjustified and instead of using cement and ash, it is possible to use road binder based on clinker and fluidized ashes in a ready mixture.

I have a request for information on how many samples for each of the 16 series were tested? Additionally, please provide the cost of soil modification.

Author Response

In my opinion, the use of polypropylene fiber is economically unjustified and instead of using cement and ash, it is possible to use road binder based on clinker and fluidized ashes in a ready mixture. I have a request for information on how many samples for each of the 16 series were tested? Additionally, please provide the cost of soil modification.

We are really appreciated for your recognition of this study. To be honest, based on the economically -friendly, environmental, and effective principles , polypropylene fibers were employed for one of the modifying material to reinforce silty soil. As for UCS tests, a total of 16×6 specimens were tested, while 16×9 specimens for triaxial tests. For all of theses tests, besides the soil transportation and equipment use expenses, modified materials cost almost ¥ 100.

Reviewer 3 Report

Dear Author,

The present study gives an insight of Mechanical properties, curing mechanism and microscopic experimental study of polypropylene fiber using fly ash modified cement-silty soil. Kindly improve the English setense . all figures and table give information and cited others research work. Paper canbe accepted with minor correction.

Author Response

Dear Author,

The present study gives an insight of Mechanical properties, curing mechanism and microscopic experimental study of polypropylene fiber using fly ash modified cement-silty soil. Kindly improve the English setense . all figures and table give information and cited others research work. Paper canbe accepted with minor correction.

Response: We are really appreciated for your recognition of this study. The language and some of the additional contents were modified to be better for readers.

Round 2

Reviewer 1 Report

ok

Author Response

Dear reviewer,

Thank you so much for your reviewing! We deeply appreciate your recognition of our research work. We wish you all the best!